# How to Compare Psychometric Factor and Network Models

**DOI:** 10.3390/jintelligence8040035

**Published:** 2020-10-02

**Authors:** Kees-Jan Kan, Hannelies de Jonge, Han L. J. van der Maas, Stephen Z. Levine, Sacha Epskamp

**Affiliations:** 1Research Institute of Child Development and Education, University of Amsterdam, 1018 WS Amsterdam, The Netherlands; h.dejonge@uva.nl; 2Department of Psychology, University of Amsterdam, 1018 WS Amsterdam, The Netherlands; h.l.j.vandermaas@uva.nl (H.L.J.v.d.M.); s.epskamp@uva.nl (S.E.); 3Department of Community Mental Health, University of Haifa, Haifa 3498838, Israel; slevine@univ.haifa.ac.il

**Keywords:** psychometric network analysis, factor analysis, latent variable modeling, intelligence, model comparison, replicating networks

## Abstract

In memory of Dr. Dennis John McFarland, who passed away recently, our objective is to continue his efforts to compare psychometric networks and latent variable models statistically. We do so by providing a commentary on his latest work, which he encouraged us to write, shortly before his death. We first discuss the statistical procedure McFarland used, which involved structural equation modeling (SEM) in standard SEM software. Next, we evaluate the penta-factor model of intelligence. We conclude that (1) standard SEM software is not suitable for the comparison of psychometric networks with latent variable models, and (2) the penta-factor model of intelligence is only of limited value, as it is nonidentified. We conclude with a reanalysis of the Wechlser Adult Intelligence Scale data McFarland discussed and illustrate how network and latent variable models can be compared using the recently developed R package Psychonetrics. Of substantive theoretical interest, the results support a network interpretation of general intelligence. A novel empirical finding is that networks of intelligence replicate over standardization samples.

## 1. Introduction

On 29 April 2020, the prolific researcher Dr. Dennis John McFarland sadly passed away. The *Journal of Intelligence* published one of his last contributions: “The Effects of Using Partial or Uncorrected Correlation Matrices When Comparing Network and Latent Variable Models” ([19]). We here provide a commentary on this paper, which McFarland encouraged us to write after personal communication, shortly before his death. The aim of our commentary is threefold. The first is to discuss the statistical procedures [19] ([19]) used to compare network and latent variable models; the second to evaluate the “penta-factor model” of intelligence; and the third to provide a reanalysis of two data sets [19] ([19]) discussed. With this reanalysis, we illustrate how we advance the comparison of network and latent variable models using the recently developed R package Psychonetrics ([6]). Before we do so, we first express our gratitude to Dr. McFarland for sharing his data, codes, output files, and thoughts shortly before he passed away, as these were insightful in clarifying what had been done precisely, and so helped us write this commentary.

As [19] ([19]) pointed out, psychometric network modeling (of Gaussian data, e.g., ([8])) and latent variable modeling ([16]) are alternatives to each other, in the sense that both can be applied to describe or explain the variance–covariance structure of observed variables of interest. As he continued, psychometric network models are typically depicted graphically, where so-called “nodes” represent the variables in the model and—depending on the choice of the researcher—”edges” represent either zero-order correlations or partial correlations among these variables. It is indeed recommended ([7]) to opt for partial correlations and also to have removed those edges that are considered not different from zero or are deemed spurious (false positives), as [19] ([19]) mentioned. This procedure of removing edges is commonly referred to as “pruning” ([7]). We would have no difficulty with the position that probably most of the psychometric network models of Gaussian data can be interpreted as “reduced” (i.e., pruned) partial correlation matrices. This also holds for the network models of intelligence that [19] ([19]) discussed ([15]; [25]).

Through personal communication, and as his R script revealed, it became clear that, unfortunately, [19] ([19]) mistook how network models of Gaussian data are produced with the R function *glasso* ([10]) and, as a consequence, how fit statistics are usually obtained. Although we are unable to verify any longer, we believe that this explains [19]’s ([19]) concern that in latent variable and psychometric network analyses “disparate correlation matrices” are being compared, i.e., zero-order correlation matrices versus partial correlation matrices. To clarify to the audience, what happens in psychometric network analysis of Gaussian data ([8]) is that two variance–covariance matrices are being compared, (1) the variance–covariance matrix implied by the network (∑) versus (2) the variance–covariance matrix that was observed in the sample (S).[note 1] This procedure parallels the procedure in structural equation modeling ([16]) in which the variance–covariance matrix implied by a latent variable model is compared to the observed sample variance–covariance matrix. In terms of *standardized* matrices, a comparison is thus always between the *zero-order correlation matrix implied by the model* and the *observed zero-order correlation matrix,* no matter whether a latent variable model is fitted in SEM software or a psychometric network model in software that allows for psychometric network analysis. This also applies when the partial correlation matrix is pruned (and ∑ is modeled as ∑ = Δ(I − Ω)^−1^Δ, where Ω contains [significant] partial correlations on the off diagonals, Δ is a weight matrix, and I is an identity matrix; see ([8])). We concur with [19]’s ([19]) statement that “in order to have a meaningful comparison of models, it is necessary that they be evaluated using identical data (i.e., correlation matrices)”, but thus emphasize that this is standard procedure.

The procedure [19] ([19]) used to compare psychometric network modes and latent variable models deviated from this standard procedure in several aspects. Conceptually, his aim was as follows:Consider a Gaussian graphical model (GGM) and one or more factor models as being competing models of intelligence.Fit both type of models (in standard SEM software) on the zero-order correlation matrix.
○Obtain the fit statistics of the models (as reported by this software).○Compare these statistics.Fit both types of models (in standard SEM software) on the (pruned) partial correlation matrix derived from an exploratory network analysis.
○Obtain the fit statistics of the models (as reported by this software).○Compare these statistics.

Applying this procedure on Wechlser Adult Scale of Intelligence ([28]) data, [19] ([19]) concluded that although the GGM fitted the (pruned) partial correlation matrix better, the factor models fitted the zero-order correlation matrix better.

We strongly advise against using this procedure as it will result in uninterpretable results and invalid fit statistics for a number of models. For instance, when modeling the zero-order correlation matrix, the result is invalid in the case of a GGM (i.e., a pruned partial correlation matrix is fitted): To specify the relations between observed variables in the software as covariances (standardized: zero-order correlations), while these relations in the GGM represent partial correlations, is a misrepresentation of that network model. Since the fit obtained from a misspecified model must be considered invalid, the fit statistics of the psychometric network model in [19]’s ([19]) Table 1 need to be discarded.

In the case [19] ([19]) modeled partial correlations, more technical detail is required. This part of the procedure was as follows:Transform the zero-order correlation to a partial correlation matrix.Use this partial correlation matrix as the input of the R *glasso* function.Take the R *glasso* output matrix as the input matrix in the standard SEM software (SAS CALIS in ([19]; [23])).Within this software, specify the relations between the variables according to the network and factor models, respectively.Obtain the fit statistics of the models as reported by this software.

This will also produce invalid statistics and for multiple reasons. Firstly, the R *glasso* function expects a variance–covariance matrix or zero-order correlation as the input, after which it will return a (sparse) inverse covariance matrix. When a partial correlation matrix is used as the input, the result is an inverse matrix of the partial correlation matrix. Such a matrix is not a partial correlation matrix (nor is it a zero-order correlation matrix), hence it is not the matrix that was intended to be analyzed. In addition, even if this first step would be corrected, meaning a zero-order correlation matrix would be used as the input of R *glasso*, such that the output would be an inverse covariance matrix (standardized: a (pruned) partial correlation matrix), the further analysis would still go awry. For example, standard SEM software programs, including SAS CALIS, expect as the input either a raw data matrix or a sample variance–covariance matrix (standardized: zero-correlation matrix). Such software does not possess the means to compare partial correlation matrices or inverse covariance matrices, and is, therefore, unable to produce the likelihoods of these matrices. In short, using the output of R *glasso* as the input of SAS CALIS leads to uninterpretable results because the likelihood computed by the software is incomparable to the likelihood of the data; all fit statistics in [19]’s ([19]) Table 2 are also invalid.

Only the SEM results of the factor models in [19]’s ([19]) Table 1 are valid, except for the penta-factor model, as addressed below. Before we turn to this model, and speaking about model fit, we deem it appropriate to first address the role of fit in choosing between competing statistical models and so between competing theories. This role is not to *lead* but to *assist* the researcher ([16]). If the analytic aim is to explain the data (rather than summarize them), any model under consideration needs to be in line with the theories being put to the test. This formed a reason as to why—in our effort to compare the mutualism approach to general intelligence with *g* theory—we ([15]) did not consider fitting a bifactor model. Bifactor models may look attractive but are problematic for both statistical and conceptual reasons (see [3]; [11]). Decisive for our consideration not to fit a bifactor model was that such a model cannot be considered in line with *g* theory ([11]; [12]), among others, because essential measurement assumptions are violated ([11]).

Note that this decision was not intended to convey that the bifactor model cannot be useful at all, theoretically or statistically. Consider the following example. Suppose a network model is the true underlying data generating mechanism and that the edges of this network are primarily positive, then one of the implications would be the presence of a positive manifold of correlations. The presence of this manifold makes it, in principle, possible to decompose the variance in any of the network’s variables into the following *variance components*: (1) a general component, (2) a unique component, and (3) components that are neither general nor unique (denoting variance that is shared with some but not all variables). A bifactor model may then provide a satisfactory *statistical summary* of these data. Yet, if so, it would not represent the data-generating mechanism. The model would *describe* but not *explain* the variance–covariance structure of the data.

The “penta-factor model” [19] ([19]) considered, which is depicted in Figure 1, is an extension of the bifactor model (see also [18]). Rather than one general factor, it includes multiple (i.e., four) general factors. As a conceptual model, we would have the same reservations as with the bifactor model.[note 2] As a statistical model, we argue against it, even when intended as a means to carry out a variance decomposition.

Firstly, a confirmatory factor model containing more than one general factor is not identified and produces invalid fit statistics (if at all the software reports fit statistics, rather than a warning or error). To solve the nonidentification, one would need to apply additional constraints, as is the case in exploratory factor modeling. Although SEM software other than SAS CALIS ([22]; [13]) was able to detect the nonidentification of the penta-factor model, the SAS CALIS software did not give an explicit warning or error when rerunning McFarland’s scripts. The software did reveal problems that relate to it, however. For instance, the standard errors of (specific) parameters could not be computed. This is symptomatic of a nonidentified model.

Secondly, even if we would have accepted the standard errors and fit measures that were reported as correct, the pattern of parameter values themselves indicate that the model is not what it seems (see Table 1, which summarizes McFarland 2020′s results concerning the WAIS-IV data of 20–54 years old). Inspection of the factor loadings reveals that only one of the four factors that were specified as general—namely factor *g*_4_—turns out to be actually general (the loadings on this factor are generally (all but one) significant). Factor *g*_3_ turns out to be a Speed factor (only the loadings of the Speed tests are substantial and statistically significant). As a result, the theoretical Speed factor that was also included in the model became redundant (none of the loadings on this factor are substantial or significant). Other factors are also redundant, including factor *g*_1_ (none of its loadings are substantial or significant). Furthermore, only one of the subtests has a significant loading on *g*_2_, which makes this factor specific rather than general.

Despite our conclusion that part of [19]’s ([19]) results are invalid, we regard his efforts as seminal and inspiring, as his paper belongs to the first studies that aim to provide a solution to the issue of how to compare network and latent variable models statistically. The question becomes: Is there a valid alternative to compare psychometric network and latent variable models? In order to answer this question, we return to the fact that standard SEM programs do not possess the means to fit GGMs. This shortcoming was the motivation as to why in previous work, we ([15]; [4]) turned to programming in the R software package OpenMx ([20]). This package allows for user-specified matrix algebraic expressions of the variance–covariance structure and is so able to produce adequate fit indices for network models. The fit statistics of these models can be directly compared to those of competing factor models. Alternatively, one could also use the recently developed and more user-friendly R package Psychonetrics ([6]). Psychonetrics has been explicitly designed to allow for both latent variable modeling and psycho-metric network modeling (and combinations thereof) and for comparing fit statistics of both types of models ([6]).

We here illustrate how a valid comparison between latent variable and network models can be conducted using this software. Codes, data, and an accompanying tutorial are available at GitHub (https://github.com/kjkan/mcfarland).

## 2. Method

### 2.1. Samples

The data included the WAIS-IV US and Hungarian standardization sample correlation matrices ([28]) that were network-analyzed by [15] ([15]) and [25] ([25]), respectively. The sample sizes were 1800 (US) and 1112 (Hungary).

### 2.2. Analysis

In R ([21]), using Psychonetrics’ (version 0.7.2) default settings, we extracted from the US standardization sample a psychometric network: That is, first the full partial correlation matrix was calculated, after which this matrix was pruned (at α = 0.01). We refer to the skeleton (adjacency matrix) of this pruned matrix as “the network model”. As it turned out, the *zero-order correlation matrix implied by the network model* fitted *the observed zero-order correlation matrix* in the (US) correlation matrix well (χ^2^ (52) = 118.59, *p* < 0.001; CFI = 1; TLI = 0.99; RMSEA = 0.026, CI_90_ = 0.20–0.33), which was to be expected since the network was derived from that matrix. One may compare this to the extraction of an exploratory factor model and refitting this model back as a confirmatory model on the same sample data. Other than providing a check if a network is specified correctly or to obtain fit statistics of this model, we do not advance such a procedure (see also [15]).

Next, we fitted the network model (truly confirmatory) on the observed Hungarian sample. Such a procedure is comparable to the cross-validation of a factor model (or to a test for configural measurement invariance). Apart from the network model, we also fitted on the Hungarian sample matrix the factor models [19] ([19]) considered, i.e., the WAIS-IV measurement model (Figure 5.2 in the US manual ([28])), a second-order *g* factor model, which explains the variance–covariance structure among the latent variables being measured, and the bifactor model. Due to its nonidentification (see above), the penta-factor model was not taken into consideration.

The network and factor models were evaluated according to their absolute and relative fit and following standard evaluation criteria ([24]): RMSEA values ≤ 0.05 (and TLI and CFI values > 0.97) were considered to indicate a good absolute fit, RMSEA values between 0.05 and 0.08 as an adequate fit, and RMSEA values between 0.08 and 0.10 as mediocre. RMSEA values >0.10 were considered unacceptable. TLI and CFI values between 0.95 and 0.97 were considered acceptable. The model with the lowest AIC and BIC values was considered to provide the best relative fit, hence as providing the best summary of the data.

The comparison between the second-order *g* factor and the psychometric network models was considered to deliver an answer to the following question. Which among these two models provides the best theoretical explanation of the variance–covariance structure among the observed variables? Is it *g* theory ([12]) or a network approach towards intelligence (consistent, for example, with the mutualism theory ([27]))?

## 3. Results

The absolute and relative fit measures of the models are shown in Table 2. From this table, one can observe the following. The absolute fit of the network model was good, meaning that the network model extracted from the US sample replicated in the Hungarian sample. In addition, both the AIC and BIC values indicated a better relative fit for the psychometric network than the *g*-factor model. The same conclusion may be derived from the RMSEA values and their nonoverlapping 90% confidence intervals. According to the criteria mentioned above, these results would thus favor the network approach over *g* theory.

The AIC and BIC values also suggested that among all models considered, the network model provided the best statistical summary, as it also outperformed the measurement model and the bifactor model. A graphical representation of the network model is given in Figure 2.

## 4. Conclusions

We contribute two points regarding the comparison of latent variable and psychometric network models. First, we pointed out that the penta-factor model is nonidentified and is therefore of only limited value as a model of general intelligence. Second, we proposed an alternative (and correct) method for comparing psychometric networks with latent variable models. If researchers want to compare network and latent variable models, we advance the use of the R package Psychonetrics ([5], [6]). Potential alternative ways we have not evaluated in this commentary may exist or are under development. These alternatives may include the recently developed partial correlation likelihood test ([26]), for example.

The results of our (re)analysis are of interest in their own right, especially from a substantive theoretical perspective. Although one may still question to what extent the psychometric network model fitted better than factor models that were not considered here, the excellent fit for the WAIS network is definitely in line with a network interpretation of intelligence ([27]). The novel finding that psychometric network models of intelligence replicated over samples is also of importance, especially in view of the question of whether or not psychometric networks replicate in general. This question is a hotly debated topic in other fields, e.g., psychopathology (see [9]; [2]). As we illustrated here and in previous work ([15]), non-standard SEM software such as OpenMx ([20]) and Psychonetrics ([5], [6]) provides a means to investigate this topic.

## Figures and Tables

**Figure 1 jintelligence-08-00035-f001:**
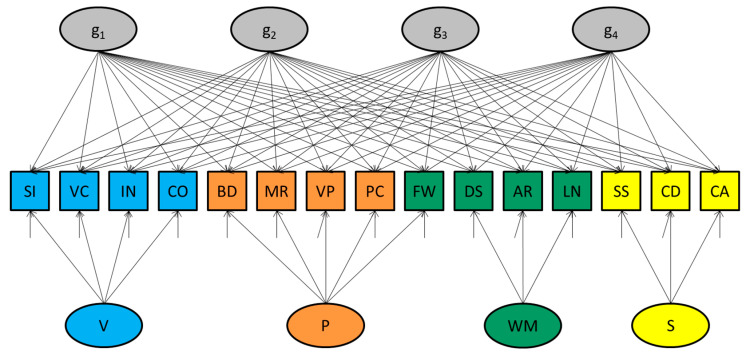
Graphical representation of the (nonidentified) penta-factor model. Note: abbreviations of subtests: SI—Similarities; VC—Vocabulary; IN—Information; CO—Comprehension; BD—Block Design; MR—Matrix Reasoning; VP—Visual Puzzles; FW—Figure Weights; PC—Picture Completion; DS—Digit Span; AR—Arithmetic; LN—Letter–Number Sequencing; SS—Symbol Search; CD—Coding; CA—Cancelation. Abbreviations of latent variables: V—Verbal factor; P—Perceptual factor; WM—Working Memory factor; S—Speed factor; *g*_1_ to *g*_4_—general factors.

**Figure 2 jintelligence-08-00035-f002:**
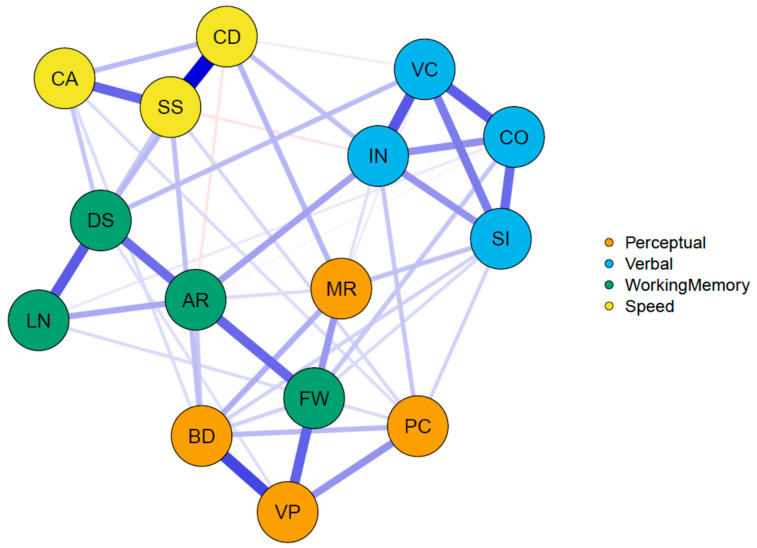
Graphical representation of the confirmatory WAIS network model. Note: abbreviations of Verbal subtests: SI—Similarities; VC—Vocabulary; IN—Information; CO—Comprehension; Perceptual subtests: BD—Block Design; MR—Matrix Reasoning; VP—Visual Puzzles; PC—Picture Completion; Working Memory subtests: DS—Digit Span; AR—Arithmetic; LN—Letter–Number Sequencing; FW—Figure Weights; and Speed subtests: SS—Symbol Search; CD—Coding; CA—Cancelation.

**Table 1 jintelligence-08-00035-t001:** [19]’s ([19]) results of the penta-factor model fitted on the WAIS-IV US standardization data of 20–54 years old.

	***Factor***		***Verbal***			***Perceptual***			***Working***			***Processing***			**Error**		
			***Comprehension***			***Organization***			***Memory***			***Speed***			**Variance**		
	**Subtest (Abbreviation)**		**Est.**	**SE**	**sig.**	**Est.**	**SE**	**sig.**	**Est.**	**SE**	**sig.**	**Est.**	**SE**	**sig.**	**Est.**	**SE**	**sig.**
1	Similarities	(SI)	−0.3413	0.0998	***										0.2928	0.0198	***
2	Vocabulary	(VC)	−0.4869	0.1317	***										0.1847	0.0243	***
3	Information	(IN)	−0.7114	0.2994	*										0.0001	^1^	
4	Comprehension	(CO)	−0.3553	0.1066	***										0.2635	0.0203	***
5	Block Design	(BD)				0.5063	0.0578	***							0.3003	0.0480	***
6	Matrix Reasoning	(MR)				0.0842	0.0575								0.4388	0.0312	***
7	Visual Puzzles	(VP)				0.4316	0.0584	***							0.3493	0.0346	***
8	Figure Weights	(FW)				0.1238	0.0507	*							0.3858	0.0255	***
9	Picture Completion	(PC)				0.2638	0.0532	***							0.6145	0.0331	***
10	Digit Span	(DS)							−0.3849	0.2596					0.3256	0.2446	
11	Arithmetic	(AR)							0.1040	0.2586					0.0001	^1^	
12	Letter–Number Sequencing	(LN)							−0.5303	0.4294					0.2344	0.4409	
13	Symbol Search	(SS)										−0.6189	0.8171		0.0001	^1^	
14	Coding	(CD)										−0.2186	0.1188		0.5036	0.1391	***
15	Cancelation	(CA)										0.1866	0.5088		0.4025	0.3731	
	***Factor***		***General Factor***			***General Factor***			***General Factor***			***General Factor***					
			***g_1_***			***g_2_***			***g_3_***			***g_4_***					
	**Subtest (Abbreviation)**		**Est.**	**SE**	**sig.**	**Est.**	**SE**	**sig.**	**Est.**	**SE**	**sig.**	**Est.**	**SE**	**sig.**			
1	Similarities	(SI)	0.1584	1.9584		0.0596	2.7983		−0.2318	0.5437		0.7130	0.1692	***			
2	Vocabulary	(VC)	0.0951	3.0534		0.1458	1.7943		−0.2079	0.1715		0.7104	0.1032	***			
3	Information	(IN)	−0.1982	4.0028		−0.2636	2.9837		−0.0322	0.4997		0.6197	0.1529	***			
4	Comprehension	(CO)	0.1793	2.2841		0.0725	3.1929		−0.2698	0.6062		0.7072	0.1914	***			
5	Block Design	(BD)	0.0198	1.8986		−0.1006	0.2779		0.0472	0.3089		0.6562	0.0574	***			
6	Matrix Reasoning	(MR)	0.0652	3.0642		−0.2035	1.0725		−0.0564	0.6956		0.7108	0.1338	*			
7	Visual Puzzles	(VP)	0.0474	2.5845		−0.1580	0.6918		0.0039	0.5324		0.6612	0.0999	***			
8	Figure Weights	(FW)	−0.0264	2.4213		−0.1714	0.3925		−0.1174	0.2968		0.7450	0.0591	***			
9	Picture Completion	(PC)	0.1317	2.1678		−0.0885	1.5528		0.1853	0.7582		0.5064	0.1370	***			
10	Digit Span	(DS)	−0.0857	1.8393		0.1220	1.3714		0.0011	0.6273		0.7099	0.1222	***			
11	Arithmetic	(AR)	−0.5003	3.5156		0.2038	7.3568		−0.1221	2.6049		0.8260	0.5251				
12	Letter–Number Sequencing	(LN)	−0.1229	1.0698		0.0517	1.8421		−0.0398	0.6581		0.6819	0.1310	***			
13	Symbol Search	(SS)	0.1119	0.9418		0.1328	0.5802		0.5088	0.1748	**	0.5726	0.0128	***			
14	Coding	(CD)	0.0763	1.2949		0.1579	0.5028		0.2938	0.0066	***	0.5758	0.0129	***			
15	Cancelation	(CA)	0.0828	1.1700		0.1277	0.0029	***	0.5838	0.0131	***	0.4458	0.0100	***			

Note: ^1^ not computed; * significant at α = 0.05; ** significant at α = 0.01; *** significant at α = 0.001.

**Table 2 jintelligence-08-00035-t002:** Results of the model comparisons based on analyses in software package Psychonetrics.

Model	χ^2^ (df)	*p*-Value	CFI	TLI	RMSEA [CI_90_]	AIC	BIC
**Network**	**129.96 (52)**	**<0.001**	**0.99**	**0.99**	**0.037 [0.029–0.045]**	**36,848.56**	**37,189.51**
Bifactor	263.41 (75)	<0.001	0.98	0.97	0.048 [0.041–0.054]	36,966.01	37,266.85
Measurement	369.47 (82)	<0.001	0.96	0.97	0.056 [0.050–0.062]	37,058.07	37,323.81
Hierarchical *g*	376.56 (84)	<0.001	0.97	0.96	0.056 [0.050–0.062]	37,061.16	37,316.87

Note: preferred model in bold.

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
