# Peer review of "How to Compare Psychometric Factor and Network Models"

_jintelligence, 2020, doi:10.3390/jintelligence8040035_

Round 1

Reviewer 1 Report

Typo:

page 3 line 134:'this issue'.

In Figure 2: See differences in Figure and Notes! Matrix reasoning is abbreviated as 'MR' instead of 'MA' and Vobabualry is abbreviated as 'VC' instead of 'VO'.

Author Response

Typo:

page 3 line 134:'this issue'.

In Figure 2: See differences in Figure and Notes! Matrix reasoning is abbreviated as 'MR' instead of 'MA' and Vobabualry is abbreviated as 'VC' instead of 'VO'.

  • Both fixed

Reviewer 2 Report

The current paper is a reply to the manuscript titled "The Effects of Using Partial or Uncorrected Correlation Matrices When Comparing Network and Latent Variable Models", published on this journal by McFarland (2020). The authors point that the analytical strategy used by McFarland is incorrect and should be avoided at all costs for it yield incorrect fit statistics. Furthermore, McFarland (2020) used a partial correlation matrix as the input to the GLASSO function, which produced an inverse matrix of the partial correlation matrix. The authors of the current paper are, basically, trying to fix the issues of the paper published by McFarland (2020). 

I cannot emphasize enough how important the current paper is, not just to respond to McFarland (2020), but to serve as a guide for future applied researchers with little experience/training in the new area combining network models and SEM.

Author Response

The current paper is a reply to the manuscript titled "The Effects of Using Partial or Uncorrected Correlation Matrices When Comparing Network and Latent Variable Models", published on this journal by McFarland (2020). The authors point that the analytical strategy used by McFarland is incorrect and should be avoided at all costs for it yield incorrect fit statistics. Furthermore, McFarland (2020) used a partial correlation matrix as the input to the GLASSO function, which produced an inverse matrix of the partial correlation matrix. The authors of the current paper are, basically, trying to fix the issues of the paper published by McFarland (2020).

I cannot emphasize enough how important the current paper is, not just to respond to McFarland (2020), but to serve as a guide for future applied researchers with little experience/training in the new area combining network models and SEM.

  • We thank Reviewer #2 for this positive assessment.

Reviewer 3 Report

In the current manuscript, the authors comment on McFarland (2020). That is, they argue against the use of certain analytic procedures and test different theories of intelligence against each other using the WAIS-IV standardization data sets of the US and Hungary. One the one hand, I think that the comment is a very helpful clarification of multiple steps taken in McFarland (2020) and provides interesting insights for theorizing about as well as testing theories of psychological constructs in general and intelligence in particular. Moreover, I very much appreciate that the authors put their data and code on GitHub in such a way that one can easily reproduce the results. On the other hand, I think that there are some minor points that could be addressed. I elaborate on these points below.

  1. I have the impression that, in the beginning, the manuscript is written in a somewhat aggressive tone, even if this aggressiveness is rather subliminal. I personally think that sentences such as “McFarland (2020) took an unfortunate turn” are not necessary. Moreover, also evaluative sentences such as “McFarland (2020) started off right” are not necessary. There is no need to judge some statement as absolutely right or wrong. It would suffice to say that the authors agree or disagree with a certain statement made in McFarland (2020) for reason x, y, z, which is a strategy they adopt later. I recommend toning down the somewhat aggressive tone and rewrite evaluative sentences.
  2. In the list of bullet points on p.2, a bullet point is missing.
  3. I admittedly did not really understand why the bi-factor model and the penta-factor model are discarded as theoretical models of intelligence on p.3. The argument is: (a) models need to be in line with theories being put to test. (b) The bi-factor model is not in line with g theory. (c) Hence, it can only summarize data (in terms of a variance decomposition) but cannot explain the data. I agree with this line of reasoning. However, from my reading of McFarland (2020) the penta-factor model was described as an operationalization of a theory, namely “the general form of the penta-factor model in the present study is consistent with Kovacs and Conway (2016) [sic] conception of multiple overlapping domain-general executive cognitive processes being required for the performance of any given test item” (p.5). Note, however, that I am neither familiar with research on intelligence in general nor these theories in particular. But if this statement in McFarland (2020) is correct, I would recommend rephrasing the section on the bi-factor model and the penta-factor model. Note that this rephrasing should not be done, in my opinion, for the statistical problems of the models or the eventual results of the model fitting but only for the theoretical status of the models.
  4. The authors write on p.7 that the GitHub page contains a tutorial. However, as far as I can see, there are data sets and the R Code to reproduce the analysis (which are very helpful) but there is no tutorial. Or do I misunderstand something?
  5. The authors could additionally recommend other ways to test the g theory and the network theory against each other in future research. For instance, the PCL Test proposed by Van Bork et al. (2019) could be used. Simulations support that it may also extend to hierarchical factor models, even if this is not (yet) mathematically proven (Van Bork, personal communication).

McFarland, D. (2020). The effects of using partial or uncorrected correlation matrices when comparing network and latent variable models. Journal of Intelligence, 8, 7. http://doi.org/10.3390/jintelligence8010007

Van Bork, R., Rhemtulla, M., Waldorp, L. J., Kruis, J., Rezvanifar, S., & Borsboom, D. (2019). Latent variable models and networks: Statistical equivalence and testability. Multivariate Behavioral Research, 1–24. http://doi.org/10.1080/00273171.2019.1672515

Author Response

In the current manuscript, the authors comment on McFarland (2020). That is, they argue against the use of certain analytic procedures and test different theories of intelligence against each other using the WAIS-IV standardization data sets of the US and Hungary. One the one hand, I think that the comment is a very helpful clarification of multiple steps taken in McFarland (2020) and provides interesting insights for theorizing about as well as testing theories of psychological constructs in general and intelligence in particular. Moreover, I very much appreciate that the authors put their data and code on GitHub in such a way that one can easily reproduce the results. On the other hand, I think that there are some minor points that could be addressed. I elaborate on these points below. 

  1. I have the impression that, in the beginning, the manuscript is written in a somewhat aggressive tone, even if this aggressiveness is rather subliminal. I personally think that sentences such as “McFarland (2020) took an unfortunate turn” are not necessary. Moreover, also evaluative sentences such as “McFarland (2020) started off right” are not necessary. There is no need to judge some statement as absolutely right or wrong. It would suffice to say that the authors agree or disagree with a certain statement made in McFarland (2020) for reason x, y, z, which is a strategy they adopt later. I recommend toning down the somewhat aggressive tone and rewrite evaluative sentences. 
  • We thank the reviewer for pointing out that the tone of the manuscript came across as somewhat aggressive. We agree. Therefore we revised our manuscript and softened the tone considerably.
  • The truth is we are specifically thankful to Dr. McFarland, who encouraged us to write our commentary. We acknowledge this in the revised manuscript.  
  1. In the list of bullet points on p.2, a bullet point is missing. 
  • Fixed 
  1. I admittedly did not really understand why the bi-factor model and the penta-factor model are discarded as theoretical models of intelligence on p.3. The argument is: (a) models need to be in line with theories being put to test. (b) The bi-factor model is not in line with g theory. (c) Hence, it can only summarize data (in terms of a variance decomposition) but cannot explain the data. I agree with this line of reasoning. However, from my reading of McFarland (2020) the penta-factor model was described as an operationalization of a theory, namely “the general form of the penta-factor model in the present study is consistent with Kovacs and Conway (2016) [sic] conception of multiple overlapping domain-general executive cognitive processes being required for the performance of any given test item” (p.5). Note, however, that I am neither familiar with research on intelligence in general nor these theories in particular. But if this statement in McFarland (2020) is correct, I would recommend rephrasing the section on the bi-factor model and the penta-factor model. Note that this rephrasing should not be done, in my opinion, for the statistical problems of the models or the eventual results of the model fitting but only for the theoretical status of the models. 
  • We agree with the reviewer that theoretical status of the models deserved more attention.
  • We rephrased the section on the bifactor model and the penta-factor model. We also added a note on how – in our reading – McFarland used the factor models to compare different theories, i.e. g theory and process overlap theory. 
  1. The authors write on p.7 that the GitHub page contains a tutorial. However, as far as I can see, there are data sets and the R Code to reproduce the analysis (which are very helpful) but there is no tutorial. Or do I misunderstand something? 
  • Thank you for pointing out our omission. The tutorial was under construction, but is now made available online.
  1. The authors could additionally recommend other ways to test the g theory and the network theory against each other in future research. For instance, the PCL Test proposed by Van Bork et al. (2019) could be used. Simulations support that it may also extend to hierarchical factor models, even if this is not (yet) mathematically proven (Van Bork, personal communication). 
  • We thank the reviewer for bringing this paper to our attention. We refer to it (in the conclusion section) and include it as a paper that addresses the question of how to compare latent variable models and network models. (Yet we consider a thorough discussion is beyond the scope of a commentary on McFarland’s paper – we hope you agree with this view).  

McFarland, D. (2020). The effects of using partial or uncorrected correlation matrices when comparing network and latent variable models. Journal of Intelligence8, 7. http://doi.org/10.3390/jintelligence8010007

Van Bork, R., Rhemtulla, M., Waldorp, L. J., Kruis, J., Rezvanifar, S., & Borsboom, D. (2019). Latent variable models and networks: Statistical equivalence and testability. Multivariate Behavioral Research, 1–24. http://doi.org/10.1080/00273171.2019.1672515

Reviewer 4 Report

The manuscript is a constructive discussion of the paper of McFarland (2020). From the authors’ point of view, some errors in the McFarland’s statistical approach are described and corrected, which are relevant and informative for the application of network analysis in general. I only have a few small issues and recommendations that might be considered.

  1. My main recommendation would be to (re-)check with the author (McFarland) beforehand to see if any points have been misunderstood of misinterpreted. In my experience, this would otherwise quickly lead to an exchange of arguments about which points the respective authors have not understood correctly (distributed over commentaries and replies). After reading both papers, I do not think that this would results in major changes in the present manuscript. However, it helps to limit oneself to the relevant issues here and, thus, make the manuscript as informative as possible.
  2. One point which is very central in the argumentation of McFarland is the use of partial correlation matrices in network analysis and uncorrected correlations in latent variable models. Although the authors have discussed this point from a psychometric perspective (e.g., inputs for the R packages), it might be useful to explicitly take up this point again from a conceptual perspective. On reading it, one gets the impression that this distinction is based on a misunderstanding of how the statistical procedures (and R packets) handle the data. However, it could be made clearer whether the argument concerning the different correlation matrices is (conceptually) relevant or is just based on a misunderstanding.
  3. “GGF fit routine” (line 90, page 2) should be explained. Since the argument is repeated again in line 99 (page 3), the specific consequence could also be briefly explained here. I think the information is a bit too short or presupposes too much previous knowledge.
  4. If applicable, you can also add Eid et al. (2018) to explain why the bifactor model or the penta-factor model are problematic. This could further strengthen the argument.
  5. Table 2: Please double check the p-value of the hierarchical g model. Based on the provided R script, I got a different value.
  6. If possible, the provided R syntax could be extended by the "penta-factor model" as shown in Fig. 1 and Tab. 1. Then all results presented would be transparent and reproducible.
  7. typos: “nissue” (line 134, page 3), “sincethe” (line 17, page 7)

References

Eid, M., Krumm, S., Koch, T., & Schulze, J. (2018). Bifactor Models for Predicting Criteria by General and Specific Factors: Problems of Nonidentifiability and Alternative Solutions. Journal of Intelligence, 6(3), 42. https://doi.org/10.3390/jintelligence6030042

Author Response

The manuscript is a constructive discussion of the paper of McFarland (2020). From the authors’ point of view, some errors in the McFarland’s statistical approach are described and corrected, which are relevant and informative for the application of network analysis in general. I only have a few small issues and recommendations that might be considered. 

  1. My main recommendation would be to (re-)check with the author (McFarland) beforehand to see if any points have been misunderstood of misinterpreted. In my experience, this would otherwise quickly lead to an exchange of arguments about which points the respective authors have not understood correctly (distributed over commentaries and replies). After reading both papers, I do not think that this would results in major changes in the present manuscript. However, it helps to limit oneself to the relevant issues here and, thus, make the manuscript as informative as possible. 
  • Unfortunately dr. McFarland passed away around the time we wrote our commentary, meaning we are unable to (re-)check if any points have been misunderstood of misinterpreted. We can say that shortly before his death, we had a quite extensive email discussion, so we believe we did not generally misinterpret his writing. Yet wherever we are unsure if our interpretation of McFarland’s intentions is correct, we mention this. 
  1. One point which is very central in the argumentation of McFarland is the use of partial correlation matrices in network analysis and uncorrected correlations in latent variable models. Although the authors have discussed this point from a psychometric perspective (e.g., inputs for the R packages), it might be useful to explicitly take up this point again from a conceptual perspective. On reading it, one gets the impression that this distinction is based on a misunderstanding of how the statistical procedures (and R packets) handle the data. However, it could be made clearer whether the argument concerning the different correlation matrices is (conceptually) relevant or is just based on a misunderstanding. 
  • We agree that a discussion on a more conceptual level may be warranted. We now better  clarified that if a partial correlation matrix is the objective of pruning, it is the zero-order correlation matrix implied by that pruned partial correlation matrix that is being compared to the observed zero-order correlation.
  • Note: McFarland’s misunderstanding was that this was not the case: “From the description of the methods by Kan et al. (2019) and Schmank et al. (2019), it is not clear what type of correlations were used to compare network and latent variable models. However, from the R code provided by Schmank et al. (2019), it is clear that the fit of network models to partial correlations were compared to the fit of latent variable models to uncorrected correlation matrices. [bold script added for emphasis]”. This is incorrect. The R code provided by Schmank et al. (2019), which is a slight adaptation of ours, clearly shows that the fits of network (and factor) models always concern fits on zero-order correlations (1. Either the zero-order correlation matrix implied by the partial correlation model or the zero-order correlation matrix implied by the latent variable model, ∑, with 2. The observed zero-order correlation matrix, S).     
  • We also included a footnote where we refer to the code of Kan et al. and Schmank et al. to clarify which matrices were compared.
  1. “GGF fit routine” (line 90, page 2) should be explained. Since the argument is repeated again in line 99 (page 3), the specific consequence could also be briefly explained here. I think the information is a bit too short or presupposes too much previous knowledge. 
  • We rephrased (used less jargon), extended, and explained more detail. 
  1. If applicable, you can also add Eid et al. (2018) to explain why the bifactor model or the penta-factor model are problematic. This could further strengthen the argument. 
  • We thank the reviewer for bringing this paper under our attention. We added the reference to our discussion of the bifactor model. 
  1. Table 2: Please double check the p-value of the hierarchical g model. Based on the provided R script, I got a different value. 
  • Checked. The p-value of the hierarchical model is correct and concerns the p-value of the absolute fit (chi-square test). We suspect the reviewer inspected the p-value of the chi-square DIFFERENCE test (which compared the g-model with the measurement model) and thus concerned a test of relative fit). We adapted our code in order to (also) show the p-values of the absolute fits. 
  1. If possible, the provided R syntax could be extended by the "penta-factor model" as shown in Fig. 1 and Tab. 1. Then all results presented would be transparent and reproducible.
  • We now provide lavaan and LISREL scripts as a separate file on GitHub, showing the model is nonidentified. In our opinion, nonidentified models should never be fitted, or their fit statistics reported (if the software would give any results).
  • Pychonetrics does not (yet) provide an error or warning that the model is unidentified. Thanks to the reviewer’s comment this is being fixed by the author of the package.
  1. typos: “nissue” (line 134, page 3), “sincethe” (line 17, page 7) 
  • Fixed

References

Eid, M., Krumm, S., Koch, T., & Schulze, J. (2018). Bifactor Models for Predicting Criteria by General and Specific Factors: Problems of Nonidentifiability and Alternative Solutions. Journal of Intelligence6(3), 42. https://doi.org/10.3390/jintelligence6030042

Reviewer 5 Report

This is a commentary to clarify some issues in the article by McFarland (2020). 

The points raised by the authors are justified and based on data, code, and outputs provided by McFarland. The paper makes a contribution to the literature by providing some clarifications and corrections, and may also be useful to researchers planning to use these methods. 

With that said, some of the conclusions are too strong, for example: 

"From a substantial theoretical perspective, the results of our analysis are clear. [...] the network interpretation of intelligence (van der Maas, et al. 2006) is to be preferred over g theory (e.g., Jensen 1998)."

I don't think that the finding RMSEA 0.037 vs. 0.056 is sufficient to make that strong a claim (what is the confidence interval of these RMSEAs? are they overlapping?). The network model fits the data slightly better, but when comparing two interpretations from a "substantial theoretical perspective" (I believe this should read "substantive"), a minor difference in model fit is only one of several points to consider. 

The paper is overall well-written, but there are some minor issues, for example: "observed sample variance-covariance matrix that was observed (p.2)", "nissue (p.3)"

Author Response

This is a commentary to clarify some issues in the article by McFarland (2020).

The points raised by the authors are justified and based on data, code, and outputs provided by McFarland. The paper makes a contribution to the literature by providing some clarifications and corrections, and may also be useful to researchers planning to use these methods.

With that said, some of the conclusions are too strong, for example:

"From a substantial theoretical perspective, the results of our analysis are clear. [...] the network interpretation of intelligence (van der Maas, et al. 2006) is to be preferred over g theory (e.g., Jensen 1998)."

  • We weakened our conclusion somewhat, also in view of your argument below.
  • “The results of our (re)analysis are of interest in their own right, especially from a substantive theoretical perspective. Although one may still question to what extent the psychometric network model fitted better than factor models that were not considered here, the excellent fit for the WAIS network is definitely in line with a network interpretation of intelligence (van der Maas, et al. 2006).”

 I don't think that the finding RMSEA 0.037 vs. 0.056 is sufficient to make that strong a claim (what is the confidence interval of these RMSEAs? are they overlapping?). The network model fits the data slightly better, but when comparing two interpretations from a "substantial theoretical perspective" (I believe this should read "substantive"), a minor difference in model fit is only one of several points to consider.

  • The RMSEA did not overlap. We added the CIS to Table 1.

 The paper is overall well-written, but there are some minor issues, for example: "observed sample variance-covariance matrix that was observed (p.2)", "nissue (p.3)" 

  • Fixed 

Reviewer 6 Report

Kan et al. submit a commentary and reply to a recently published article in the Journal of Intelligence (McFarland, 2020). Specifically, these authors present four points. First, they address the overall description of psychometric network analysis by McFarland. Next, scrutiny of the statistical procedures and the “Penta-Factor” model of intelligence used in the McFarland publication are proffered. Finally, a reanalysis of the data used by McFarland is provided using confirmatory latent variable and confirmatory psychometric network modeling. The article concludes that data collected from cognitive test batteries, like the Wechsler Adult Intelligence Scale-Fourth Edition (WAIS-IV), are more consistent with psychometric network models than latent variable models.

Although the manuscript contains several proofreading errors, the general message is well structured. The introduction logically motivates the current project, the methods and statistical methodology are well outlined and described, and the associated R code is understandable and accessible. Generally, Kan et al. provide evidence that the statistical procedure followed by McFarland for conducting psychometric network modeling was not in line with recent publications or standard practices. Additionally, Kan et al. conclude that the network interpretation of intelligence should be preferred over latent variable models like g theory and the Penta-factor model. This work demonstrates a very salient argument among individual difference researchers in cognitive psychology, what is the underlying structure of intelligence data. The current work by Kan et al. proposes that the data generating mechanism behind intelligence data is an interconnected, interactive network and not the traditional latent variable model that has been featured for over 100 years.

Furthermore, Kan et al. present their reanalysis as a “how to”-guide for making statistical comparisons of latent variable and psychometric network modeling based on best practices and current standards. I especially appreciate this perspective as psychometric network analysis continues to increase in relevance among various psychological subfields. Additionally, it was enlightening to get some details on how McFarland conducted their analyses as this was not clear in the original publication. Finally, I appreciated the careful and deliberate discussion of model fit and the role that fit has in assisting researchers interested in choosing between alternative theoretical accounts.

Overall, I found the commentary to be very well composed and concise; and in fact, I had some difficulty finding areas that need improvement. Kan et al. offer constructive criticism of the techniques used by McFarland for conducting psychometric network analysis and provide information on making direct comparisons between confirmatory latent variable modeling and confirmatory psychometric network analysis.

What follows are a list of major concerns:   

(1) Throughout the document there are several areas that require proofreading to determine whether or what is missing. I have listed areas that stuck out to us the most below:

Abstract, line 22-23, p. 1:

The words “interpretation of” are repeated at the end of line 22 and beginning of 23

Introduction, lines 59-64, p. 2:

The clarification offered about how variance-covariance matrices are compared when conducting psychometric network modeling was very strong. As network modeling becomes more popular it will be important for future researchers to understand this. I do want to bring attention towards the end of this statement on line 64, as there seems to be a potentially missing word in the phrase “…but in psychometric network analysis general”.

Introduction, line 134, p. 3:

There is a misspelling in the first phrase of the sentence, “Apart from this nissue

Introduction, line 4, p. 7:

Github is misspelled at beginning of line “at Gitub”

Method, line 17, p. 7:

Towards the end of the line the words since and the do not have a space separating them, “to be expected sincethe network was…”

Method, line 23, p. 7:

Towards the middle of this line it seems like there is a word missing, “Such procedure is comparable to the cross-validation a factor model

(2) Introduction, lines 38-40, p.1:

I appreciated the acknowledgment of Dr. Dennis McFarland, however, with his recent passing in April of 2020 I wondered if it would be appropriate to also include an acknowledgement of this as well.

(3) Method, lines 24-28, p. 7

The measurement model, second order g factor model, and the bifactor models are well known in the literature, however, would it be possible to include a statement or figure about how these models were specified? What would be particularly informative is whether cross-loadings were present in any of the latent variable models (i.e., Figure Weights being explained by the Perceptual and Working Memory factors).

(4) Figure 1, p. 4:

The two latent variables representing the Perceptual and Verbal factors appear to need to be switched. Now the diagram implies that Similarities, Information, Vocabulary, and Comparison subtests are explained by the Perceptual factor and not the Verbal factor.

(5) Table 2, p. 8:

When presenting the model fit for your assessed models if there was a reason for not including a Comparative model fit index (e.g., Tucker-Lewis Index or Comparative Fit Index)? In the past I have noticed that latent variable models of intelligence tend to have variable values of Comparative model fit indices, so it would be informative to see if that is the case in current project.

Author Response

Kan et al. submit a commentary and reply to a recently published article in the Journal of Intelligence (McFarland, 2020). Specifically, these authors present four points. First, they address the overall description of psychometric network analysis by McFarland. Next, scrutiny of the statistical procedures and the “Penta-Factor” model of intelligence used in the McFarland publication are proffered. Finally, a reanalysis of the data used by McFarland is provided using confirmatory latent variable and confirmatory psychometric network modeling. The article concludes that data collected from cognitive test batteries, like the Wechsler Adult Intelligence Scale-Fourth Edition (WAIS-IV), are more consistent with psychometric network models than latent variable models.

Although the manuscript contains several proofreading errors, the general message is well structured. The introduction logically motivates the current project, the methods and statistical methodology are well outlined and described, and the associated R code is understandable and accessible. Generally, Kan et al. provide evidence that the statistical procedure followed by McFarland for conducting psychometric network modeling was not in line with recent publications or standard practices. Additionally, Kan et al. conclude that the network interpretation of intelligence should be preferred over latent variable models like g theory and the Penta-factor model. This work demonstrates a very salient argument among individual difference researchers in cognitive psychology, what is the underlying structure of intelligence data. The current work by Kan et al. proposes that the data generating mechanism behind intelligence data is an interconnected, interactive network and not the traditional latent variable model that has been featured for over 100 years.

Furthermore, Kan et al. present their reanalysis as a “how to”-guide for making statistical comparisons of latent variable and psychometric network modeling based on best practices and current standards. I especially appreciate this perspective as psychometric network analysis continues to increase in relevance among various psychological subfields. Additionally, it was enlightening to get some details on how McFarland conducted their analyses as this was not clear in the original publication. Finally, I appreciated the careful and deliberate discussion of model fit and the role that fit has in assisting researchers interested in choosing between alternative theoretical accounts.

Overall, I found the commentary to be very well composed and concise; and in fact, I had some difficulty finding areas that need improvement. Kan et al. offer constructive criticism of the techniques used by McFarland for conducting psychometric network analysis and provide information on making direct comparisons between confirmatory latent variable modeling and confirmatory psychometric network analysis.

  • We thank the reviewer for this positive assessment.

 What follows are a list of major concerns:   

 (1) Throughout the document there are several areas that require proofreading to determine whether or what is missing. I have listed areas that stuck out to us the most below:

 Abstract, line 22-23, p. 1:

The words “interpretation of” are repeated at the end of line 22 and beginning of 23

Introduction, lines 59-64, p. 2:

The clarification offered about how variance-covariance matrices are compared when conducting psychometric network modeling was very strong. As network modeling becomes more popular it will be important for future researchers to understand this. I do want to bring attention towards the end of this statement on line 64, as there seems to be a potentially missing word in the phrase “…but in psychometric network analysis general”.

 Introduction, line 134, p. 3:

There is a misspelling in the first phrase of the sentence, “Apart from this nissue

 Introduction, line 4, p. 7:

Github is misspelled at beginning of line “at Gitub”

 Method, line 17, p. 7:

Towards the end of the line the words since and the do not have a space separating them, “to be expected sincethe network was…”

Method, line 23, p. 7:

Towards the middle of this line it seems like there is a word missing, “Such procedure is comparable to the cross-validation a factor model

  • Thank you for highlighting these errors. We have fixed them and carefully reviewed the manuscript for grammatical, spelling and typing errors.

(2) Introduction, lines 38-40, p.1:

 I appreciated the acknowledgment of Dr. Dennis McFarland, however, with his recent passing in April of 2020 I wondered if it would be appropriate to also include an acknowledgement of this as well.

  • We now present our commentary in memory of Dr. McFarland. We also thank him in the acknowledgement.

 (3) Method, lines 24-28, p. 7

The measurement model, second order factor model, and the bifactor models are well known in the literature, however, would it be possible to include a statement or figure about how these models were specified? What would be particularly informative is whether cross-loadings were present in any of the latent variable models (i.e., Figure Weights being explained by the Perceptual and Working Memory factors).

  • Because of space limitations, we refer to McFarland’s paper (and the WAIS manual).
  • We included the figures in our tutorial on GitHub though

 (4) Figure 1, p. 4:

The two latent variables representing the Perceptual and Verbal factors appear to need to be switched. Now the diagram implies that Similarities, Information, Vocabulary, and Comparison subtests are explained by the Perceptual factor and not the Verbal factor.

  • Fixed

 (5) Table 2, p. 8:

When presenting the model fit for your assessed models if there was a reason for not including a Comparative model fit index (e.g., Tucker-Lewis Index or Comparative Fit Index)? In the past I have noticed that latent variable models of intelligence tend to have variable values of Comparative model fit indices, so it would be informative to see if that is the case in current project. 

  • We agree the TLI and CL values are informative. We added them to the table.

Round 2

Reviewer 1 Report

Ready for publication!

Reviewer 3 Report

The authors addressed all my minor points to my full satisfaction. I especially applaud the authors for the extremely helpful GitHub page with all data, code, and the detailed tutorial. The manuscript further clarifies in an accessible way how to compare latent variable and network models as applied to research on intelligence.

Reviewer 4 Report

The authors have done a good job with the revision. I did not notice any other issues that would stand in the way of publication.

Reviewer 5 Report

The authors sufficiently addressed the comments raised by the reviewers. I have no remaining concerns. 

Reviewer 6 Report

A well-written and necessary clarification; my previous concerns have been mitigated